# Effect of Ni Doping Content on Phase Transition and Electrochemical Performance of TiO_2_ Nanofibers Prepared by Electrospinning Applied for Lithium-Ion Battery Anodes

**DOI:** 10.3390/ma13061302

**Published:** 2020-03-13

**Authors:** Danning Kang, Jun Li, Yuyao Zhang

**Affiliations:** School of Materials Engineering, Shanghai University of Engineering Science, Shanghai 201620, China; Kangdanning@126.com (D.K.); yuyaozhang01@163.com (Y.Z.)

**Keywords:** electrospinning, Ni doping, TiO_2_ nanofibers, phase transition, lithium ion battery

## Abstract

Titanium dioxide (TiO_2_), as a potential anode material applied for lithium-ion batteries (LIBs), is constrained due to its poor theoretical specific capacity (335 mAh·g^−1^) and low conductivity (10^−7^-10^−9^ S·cm^−1^). When compared to TiO_2_, NiO with a higher theoretical specific capacity (718 mAh·g^−1^) is regarded as an alternative dopant for improving the specific capacity of TiO_2_. The present investigations usually assemble TiO_2_ and NiO with a simple bilayer structure and without NiO that is immersed into the inner of TiO_2_, which cannot fully take advantage of NiO. Therefore, a new strategy was put forward to utilize the synergistic effect of TiO_2_ and NiO, namely doping NiO into the inner of TiO_2_. NiO-TiO_2_ was fabricated into the nanofibers with a higher specific surface area to further improve their electrochemical performance due to the transportation path being greatly shortened. NiO-TiO_2_ nanofibers are expected to replace of the commercialized anode material (graphite). In this work, a facile one-step electrospinning method, followed by annealing, was applied to synthesize the Ni-doped TiO_2_ nanofibers. The Ni doping content was proven to be a crucial factor affecting phase constituents, which further determined the electrochemical performance. When the Ni doping content was less than 3 wt.%, the contents of anatase and NiO were both increased, while the rutile content was decreased in the nanofibers. When the Ni doping content exceeded 3 wt.%, the opposite changes were observed. Hence, the optimum Ni doping content was determined as 3 wt.%, at which the highest weight fractions of anatase and NiO were obtained. Correspondingly, the obtained electronic conductivity of 4.92 × 10^−5^ S⋅cm^−1^ was also the highest, which was approximately 1.7 times that of pristine TiO_2_. The optimal electrochemical performance was also obtained. The initial discharge and charge specific capacity was 576 and 264 mAh·g^−1^ at a current density of 100 mA·g^−1^. The capacity retention reached 48% after 100 cycles, and the coulombic efficiency was about 100%. The average discharge specific capacity was 48 mAh·g^−1^ at a current density of 1000 mA·g^−1^. Approximately 65.8% of the initial discharge specific capacity was retained when the current density was recovered to 40 mA·g^−1^. These excellent electrochemical results revealed that Ni-doped TiO_2_ nanofibers could be considered to be promising anode materials for LIBs.

## 1. Introduction

There will be huge demand for rechargeable energy storage systems (particularly lithium ion batteries (LIBs)) with the fast-growing energy consumption on earth and the development of renewable energy storage devices in recent years (portable electronics (PEs) and electric vehicles (EVs) and so on). Graphite is commonly used as the anode material in commercial LIBs [1,2,3,4]. However, the electrode potential of carbon-based materials is about 0.15–0.25 V (vs Li^+^/Li), which is close to that of lithium 0 V (vs Li^+^/Li) [5]. When the battery is overcharged, metallic lithium might be precipitated on the surface of the carbon electrode to form the dendrites, causing a short circuit or a thermal runaway at high temperatures. On the other hand, the surface is subject to being covered with a solid electrolyte interface (SEI), which causes a sharp attenuation in rate capability and specific capacity [6]. Therefore, the present investigations focus on the exploration of new anode materials as an alternative for graphite.

Transition metal oxide (TiO_2_) is considered to be an ideal anode material for LIBs [7,8]. First and foremost, TiO_2_ as a kind of “zero strain” material hardly suffers from drastic volume changes during the Li insertion/extraction [9], thus demonstrating the excellent cycling stability. In addition, TiO_2_ has a higher potential of discharge platform (1.78 V vs Li^+^/Li) [10] than that of graphite, so that the formation of hazardous Li dendrites can be effectively avoided. Unfortunately, TiO_2_, as a semiconductor, has poor conductivity (10^−7^–10^−9^ S·cm^−1^), which greatly limits the enhancement in electrochemical performance with respect to specific capacity and rate capability due to the charge transportation strongly inhibited [11,12,13]. Some strategies (such as the nanomerization of TiO_2_ and the introduction of other substances into TiO_2_) have been put forward to address the above shortcomings [14,15]. A large number of researches had proved that nanocrystallization can shorten the ion migration path, and the electrochemical reaction activity of the material can be increased due to the increase in specific surface area, with respect to the nanomerization of TiO_2_ [16,17]. Zheng et al. [18] embedded TiO_2_ nanoparticles into microporous amorphous carbon spheres through pyrrole polymerization and carbonization to form a watermelon-like structure. P25@C nanoparticles retained a high specific capacity of 107 mAh·g^−1^ after 5000 cycles at a current density of 20 C. The average capacity loss rate per charge and discharge cycle was less than 0.01%. The excellent electrochemical performance was attributed to the watermelon-like composite structure, which improved the conductivity and structural stability of TiO_2_ nanoparticles by eliminating the agglomeration of TiO_2_ nanoparticles and improving the reaction efficiency. Ren et al. [19] synthesized a hollow structure TiO_2_ nanosphere, with a size of approximately 7 nm, which significantly shortened the lithium ions transmission path. The electrochemical test results showed that the hollow structure TiO_2_ nanosphere owned a higher specific capacity of 212 mAh·g^−1^ at a current density of 20 C, which was 12.5 times that of ordinary TiO_2_ anode materials. Fan et al. [20] successfully synthesized TiO_2_/graphene composites by uniformly dispersing TiO_2_ nanoparticles in the graphene sheetby a the simple sol-gel method. The first discharge specific capacity was 302 mAh·g ^−1^ at a current density of 15 mA·g^−1^, which was higher than that of pure TiO_2_ (233 mAh·g^−1^). Under the current density of 15, 75, 150, 300, 450, and 750 mA·g^−1^, the rate performance test results showed that the discharge specific capacity of TiO_2_/graphene was higher than that of pure TiO_2_. This result was explained in terms of the better conductivity of the composite material, which was conducive to lithium ions deintercalation. To date, the electrospinning method, hydrothermal method, template method, and electrochemical deposition method prepare the various forms of TiO_2_ nanotubes, nanowires, nanocolumns, and nanoparticles [21,22,23,24]. Among all the nanostructures, one-dimensional nanostructured TiO_2_ fibers have attracted the most attention due to their advantages, such as directional electron conduction, short ion-transmission paths, strong stress tolerance, and large electrochemically active surface area [25]. Electrospinning is considered to be an effective method for preparing one-dimensional nanomaterials, owing to its simple operation, controllable parameters, and mass production [26]. Panda et al. [27] prepared a three-dimensional TiO_2_ nanotube array that was tightly bonded to the substrate by anodizing Ti film and found that its specific capacity was effectively improved, along with a significant increase in the specific surface area of TiO_2_. The TiO_2_ tubes with a wall thickness of 40 nm demonstrated a discharge specific capacity of only 170 mAh·g^−1^, while the value was increased to 330 mAh·g^−1^ when the wall thickness was reduced to 5 nm. Armstrong et al. [28] prepared TiO_2_ nanowire anode materials via the hydrothermal reaction and investigated their electrochemical performance with bulk TiO_2_ as a reference. The results demonstrated that the TiO_2_ nanowires owed a higher specific capacity than bulk TiO_2_ (305 vs 240 mAh·g^−1^). 

In addition to nanomerization, the inclusion of additional substances, such as α-Fe_2_O_3_ [29], Ag [30], graphene [31], amorphous carbon [32], and Co_3_O_4_ [33], were introduced into the TiO_2_ structure was also found to be an effective method for improving the electrochemical performance. Zhang et al. [34] applied the electrospinning technology to prepare one-dimensional mesoporous Ag@TiO_2_ nanofibers. After 100 cycles at a current density of 100 mA·g^−1^, as compared to pristine TiO_2_ nanofibers (73 mAh·g^−1^), mesoporous Ag@TiO_2_ nanofibers had excellent discharge specific capacity of 128 mAh·g^−1^. In addition, the electrode also showed a better rate performance. As the current density was increased from 40 to 1000 mA·g^−1^, and finally returned to 40 mA·g^−1^, the electrode retained 80% of the initial value (about 162.25 mAh·g^−1^), while the capacity retention rate of pristine TiO_2_ was only 55.42% of the initial value. Li et al. [35] synthesized boron-doped anatase TiO_2_ nanofibers by electrospinning. At a current density of 4000 mA·g^−1^, boron-doped TiO_2_ owned a much higher specific capacity of 147 mAh·g^−1^ than pristine TiO_2_ (52 mAh·g^−1^). The discharge specific capacity remained at 168 mAh·g^−1^ after 5000 cycles at 2000 mA·g^−1^. It was found that boron-doped TiO_2_ nanofibers had excellent cycle stability under the large current densities and long cycle periods. It was attributed to the improvement in conductivity of TiO_2_ nanofibers that resulted from the addition of boron, which allowed for the high reversible capacity under the large current densities. In 2000, Tarascon et al. [36] first proposed that nano-sized transition metal oxides (Fe, Co, Mn, Ni, etc.) could be selected as anode materials for LIBs based on a conversion reaction ($\mathrm{MxOy}+2{\mathrm{yLi}}^{+}+2{\mathrm{ye}}^{-}↔\mathrm{xM}+{\mathrm{yLi}}_{2}O$). Many investigations into the transition metal-doped TiO_2_ were subsequently carried out. Fehse et al. [37] prepared Nb-doped TiO_2_ nanofibers via the facile one-step electrospinning method, and they exhibited a larger rate capability (~23 mAh·g^−1^ at 5 C) than undoped TiO_2_ (~10 mAh·g^−1^ at 5 C). Opra et al. [38] fabricated Zr-doped TiO_2_ nanotubes while using the sol-gel process. The optimal capacity of Zr-doped TiO_2_ reached 135 mAh·g^−1^ after 35 charge/discharge cycles, which was 2.7 times that of undoped TiO_2_ (50 mAh·g^−1^). The other transition metals, such as Sn [39], Hf [40], and Fe [41], had also been doped into TiO_2_ and proved to contribute to improving the electrochemical performance of TiO_2_. When doping TiO_2_ with Sn and Fe, a higher specific capacity can be obtained due to the formation of SnO_2_ and Fe_2_O_3_, which have a high theoretical specific capacity, typically 781 mAh·g^−1^ for SnO_2_ and 1005 mAh·g^−1^ for Fe_2_O_3_. It will contribute to the improvement in specific capacity of Sn/Fe doped TiO_2_. However, SnO_2_ and Fe_2_O_3_ will suffer from the drastic volume changes during the charging/discharging process, thus causing the pulverization and loss of the energy capacity [39,41,42]. Therefore, seeking a reasonable dopant into TiO_2_ has become a hot topic for researchers. Nickel oxide (NiO) can be regarded as a promising dopant to solve the above-mentioned shortcomings, owing to its high theoretical specific capacity (718 mAh·g^−1^), abundant resources, low cost, and environmental friendliness. Kyeremateng et al. [43] first prepared TiO_2_ nanotube (TiO_2_nt) layers on commercial titanium foils by electrochemical anodization, then synthesized NiO on TiO_2_nt layers in an aqueous electrolytic bath containing Ni^2+^ by electrodeposition (a current density of 2.6 mA·cm^−2^ for 120 s), followed by an annealing treatment at 500 °C under open air atmosphere. The electrochemical test results showed that the introduction of NiO caused a large increase in capacity. The initial discharge area capacity of the TiO_2_nt+NiO (170 µAh·cm^−2^) was approximately three times that of TiO_2_nt (58 µAh·cm^−2^). The value of TiO_2_nt+NiO suffering from 25 cycles at 70 µA·cm^−2^ retained 80 µAh·cm^−2^, which was significantly higher than that of TiO_2_nt (30 µAh·cm^−2^). Li et al. [44] prepared TiO_2_ nanosheet arrays on a Ti foil while using the hydrothermal method. Subsequently, NiO was deposited onto TiO_2_ immersed in aqueous ammonia, NiSO_4_·4H_2_O, and K_2_S_2_O_8_ mixed liquors to synthesize TiO_2_@NiO nanosheet arrays. Finally, the samples were annealed at 350 °C for 2 h in argon to obtain anode materials. TiO_2_@NiO nanosheet arrays retained a very high specific capacity of 376 mAh·g^−1^ after 100 cycles at the current density of 200 mA·g^−1^, while the value was only 179 mAh·g^−1^ for TiO_2_ nanosheet arrays. Zhang et al. [45] acquired titanium acid (NTA) nanotubes via the chemical reactions occurring between commercial P25-TiO_2_ and NaOH solution. Subsequently, NiO-NTA was successfully synthesized by directly dropping an ethanol solution containing Ni(NO_3_)_2_·6H_2_O onto NTA nanotubes, followed by the annealing treatment at 400 °C for 3 h. The initial discharge/charge specific capacity of NTA reached 297/256 mAh·g^−1^ at the current density of 0.5 C, while the value was improved to 203/177 mAh·g^−1^ by introducing NiO into NTA. Furthermore, the specific capacity of NTA rapidly attenuated with an increase in the current density (255, 125, 102, 79, 63, and 48 mAh·g^−1^ for 0.5 C, 1 C, 2 C, 3 C, 5 C, and 10 C). However, NiO-NTA demonstrated the best rate capability (257, 240, 216, 192, 162, and 130 mAh·g^−1^). Chen et al. [46] synthesized a TiO_2_-NiO nanoparticle precursor via the sol-gel method, and then obtained TiO_2_-NiO nanoparticles by the calcining treatment at 300 °C for 2 h. The initial capacity of the TiO_2_ electrodes was improved from 170 mAh·g^−1^ to 300 mAh·g^−1^ at 20 mA·g^−1^ by introducing NiO into TiO_2_.

All of the above-mentioned researches had confirmed that the addition of Ni contributed to the improvement in electrochemical performance of TiO_2_. TiO_2_ owns the excellent cycling stability, and NiO can provide superior discharge specific capacity. The Ni-doped TiO_2_ nanofibers may be endowed with excellent electrochemical performance due to the synergistic effect between the two, which demonstrate the potential application as a prospective anode for high-performance LIBs. However, the above-mentioned researches also have some disadvantages. First, most of the NiO-TiO_2_ nanocomposites were prepared by the two-step method, namely nanostructured TiO_2_ was firstly prepared, and then NiO was synthesized on pre-prepared TiO_2_. The separated bilayer structure resulted in the interior of TiO_2_ free of NiO, and a portion of TiO_2_ surfaces being uncovered with NiO. This might led to the synergistic effect being greatly weakened. Second, the effect of nickel concentration on electrochemical performances and the evolution in phase constituents was not investigated. As a matter of fact, it is well known that as-prepared TiO_2_ usually exists in two forms of anatase and rutile. However, anatase TiO_2_ plays a decisive role in electrochemical performance owing to its unique structure when compared to rutile TiO_2_. Therefore, it is very essential to establish a relationship among the Ni doping content, relative content of anatase and rutile, and electrochemical performance.

Electrospinning followed by annealing successfully synthesized one-dimensional NiO-TiO_2_ nanofibers in this study. The effect of the Ni doping content on phase constituents was investigated in detail. The evolution in electrochemical performance was also clearly revealed. The optimal Ni doping content was finally determined. 

## 2. Experimental 

### 2.1. Synthesis of Pristine TiO_2_ and Ni-doped TiO_2_ Nanofibers

Tetra-butyl ortho-titanate (TBOT, CP, 99%, Sinopharm Chemical Reagent Co., Ltd, Shanghai, China), absolute ethyl alcohol (GR, ≥99.7%, Sinopharm Chemical Reagent Co., Ltd, Shanghai, China), acetic acid (AR, ≥99.5%, Sinopharm Chemical Reagent Co., Ltd, Shanghai, China), polyvinylpyrrolidone (PVP, AR, M_W_ ≈ 1,300,000 g⋅mol^−1^, Aladdin Industrial Corporation, Shanghai, China), and Ni(NO_3_)_2_⋅6H_2_O (nickel(II) nitrate hexahydrate, AR, Sinopharm Chemical Reagent Co., Ltd, Shanghai, China) were used to produce pristine and Ni-doped TiO_2_ nanofibers. 0.5 g PVP was first dissolved in 8 mL absolute ethanol under the rapid stirring of 2 h to ensure PVP was completely dissolved in order to prepare pristine TiO_2_ nanofibers. Subsequently, acetic acid and TBOT in a volume ratio of 1:1 were introduced into the above solution, and continuously stirred for 6 h until the solution exhibited a uniform transparent state. For the Ni-doped TiO_2_ nanofibers (1, 3, 3.5, 4, 6, and 10 wt.%), Ni(NO_3_)_2_⋅6H_2_O corresponding to different masses and 2 mL absolute ethanol were dissolved into the above solution by stirring to form a light green solution free of precipitation. Afterwards, the above solution was transferred into a 20 mL syringe with a stainless-steel needle (18 G). During the electrospinning process, aluminum foil was applied to receive the filaments from the needle. The distance between needle point and aluminum foil was adjusted to 12 cm, and the applied voltage was maintained at 8 kV. The humidity was limited to 40 RH% to prevent hydrolysis of TBOT. Finally, the obtained precursor was put into a muffle furnace and then annealed at 600 °C for 5 h in air. After annealing, the white films (pristine TiO_2_ nanofibers) and khaki films (different Ni-doped TiO_2_ nanofibers) were obtained for the subsequent experiments.

### 2.2. Assembly of LIBs

Annealed nanofibers as the active material, polyvinylidene fluoride (PVDF) as the binder, and acetylene black as the carbon source were mixed at a mass ratio of 8:1:1; then, 1 mL N-methyl-2-pyrrolidone (NMP) was added to form the slurries. The obtained slurries were coated onto copper foils by a coating machine (MSK-AFA-III, HF-Kejing, China) and then dried at 60 °C for 15 h in a vacuum oven to obtain the TiO_2_ and Ni-doped TiO_2_ working electrodes. Lithium metal foil was selected as the counter electrode. The diaphragm was made of Celgard’s 2400 polypropylene film. The electrolyte was 1 M lithium hexafluorophosphate (LiPF_6_) that was dissolved into ethylene carbonate (EC) and diethyl carbonate (DEC) (EC:DMC=1:1, v/v); 2032 type button batteries (Figure 1) were assembled in a glove box (Super 1220/750/900, Shanghai Mikrouna Electromechanical Technology Co., Ltd, Shanghai, China) that were filled with high purity argon (purity of 99.999%).

### 2.3. Characterization

The crystal structure of the as-prepared samples was analyzed while using an X-ray diffractometer (XRD, D2-PHASER Bruker, Karlsruhe, Germany). The X-ray profiles were measured between 10 and 80°, (2θ°) with a Cu Kα_1_ irradiation source (λ = 0.1540560 nm). The applied tube current and voltage were 40 mA and 40 kV. An X-ray photoelectron spectroscope detected the elements that were involved in the samples and their chemical valence states (XPS, ESCALAB 250XI, Thermo Fisher Scientific, Waltham, MA, USA), and all of the spectra were calibrated by using C 1s adventitious carbon as a reference binding energy (284.8 eV) [34]. Raman scattering was measured using a laser Raman spectrometer (Raman, inVia Reflex/inVia Reflex) with a 633 nm excitation. The surface morphology, microstructure, the elemental mapping, and chemical compositions were investigated by a field-emission scanning electron microscope (FESEM, S-4800, Hitachi, Tokyo, Japan), with an acceleration voltage of 15 kV, and a transmission electron microscope (TEM, JEM-2100F, JOEL, Tokyo, Japan) with an acceleration voltage of 200 kV that was equipped with an energy-dispersive X-ray detector (EDS, X-MAX 65T, OXFORD, England, UK).

The electrochemical performances of Ni-doped TiO_2_ nanofibers were investigated by assembling CR2032 coincells with the lithium foil as the counter electrode. The cycle stability was measured through galvanostatic charge/discharge tests under the voltage between 0.05 and 3 V (vs Li/Li^+^) at 100 mA⋅g^−1^ for 100 cycles, and the rate capability was then evaluated at different current densities: 40, 100, 200, 400, 1000, and finally 40 mA⋅g^−1^. Six data points were acquired at each density. All of the above tests were carried out on an electrochemical workstation (CT 4008, Neware Electronics Co., Ltd, China). The electrochemical impedance spectroscopy (EIS) was tested at room temperature on an electrochemical workstation (CS 310H, China). An AC amplitude of 10 mV was set with a frequency window from 0.1 MHz to 0.01 Hz. The cyclic voltammetry (CV) test was carried out on an electrochemical workstation (CHI 760E, CH Instruments Ins, China), with the voltage range of 0–3 V (vs. Li^+^/Li) and a sweeping rate of 0.2 mV⋅s^−1^.

## 3. Results and Discussion

### 3.1. Phase and Morphology Characterization of Nanofibers

Figure 2 showed the XRD patterns of TiO_2_ nanofibers with different Ni doping contents, which were synthesized by electrospinning, followed by annealing at 600 °C. Sharp diffraction peaks were clearly observed, which indicated that all of the samples subject to annealing had a high crystallinity. Primary phases with respect to anatase (JCPDS, No. 00-021-1272) and rutile (JCPDS, No. 01-071-0650) were identified. Characteristic peaks that were related to anatase TiO_2_ could be observed at 25.5°, 37.9°, and 48.3°, accompanied by some weak peaks located at 55.3°, 62.7°, and 70.18°. Several diffraction peaks that were associated with rutile TiO_2_ appeared at 27.5°, 36.1°, 41.2°, 44.1°, 62.7°, and 64.0°. When Ni was doped into TiO_2_, traces of NiO (JCPDS, No. 03-065-2901) may be synthesized, since two weak peaks that were related to NiO were observed at 37.2° and 75.4°. When the Ni doping content exceeded 3 wt.%, three new peaks located at 24.2°, 33.1°, and 49.5° appeared and presented an increasing tendency with the increase in Ni doping content. The indexed result demonstrated that another new compound of NiTiO_3_ (JCPDS, No. 01-070-0335) was formed.

By comparing the XRD patterns with the samples with different Ni doping contents, two interesting phenomena can be observed, which corresponded to changes in position and intensity ratio of the two strongest peaks that were related to anatase TiO_2_ (101) and rutile TiO_2_ (110). With respect to the position, the two peaks gradually shifted to the left with an increase in Ni doping content from 0 to 3 wt.%, which resulted from the solid solution of Ni into the TiO_2_ lattice. When the Ni doping content exceeded 3 wt.%, the two peaks moved toward the opposite direction. This meant that the maximum solid solubility of Ni in TiO_2_ was reached when the Ni doping content reached 3 wt.%. The Rietveld refinement was applied to precisely calculate the changes in lattice constant that result from the Ni doping. Table 1 showed the relationship between the lattice constants of TiO_2_ and Ni doping content. As the doping content of Ni was increased from 0 wt.% to 3 wt.%, the lattice constants for anatase and rutile were gradually increased (the cell volume was increased from 61.97 to 62.52 for rutile, and from 136.20 to 136.42 for anatase). However, the lattice constants of TiO_2_ presented the contrary change along the further increase from 3 wt.% to 10 wt.% in the Ni doping content (the cell volume was reduced from 62.52 to 62.04 for rutile, and from 136.42 to 135.36 for anatase). The calculated results were well consistent with the observed phenomenon. The intensity ratio between (101) of anatase TiO_2_ and (110) of rutile TiO_2_ was extremely low for the sample without doping Ni. This observation can be attributed to the spontaneous and irreversible transformation of metastable anatase TiO_2_ to stable rutile TiO_2_. It was noticeable that transformation can be regarded as a structural reconstruction process, in which the bonds among the atoms will be broken and reformed by atomic diffusion. The weight fraction of anatase and rutile TiO_2_ can be calculated by the empirical formula, as follows [47]:(1)$$X_{A}=\frac{100}{\left(1+1.265\frac{I_{R}}{I_{A}}\right)}$$
where X_A_ represented the percent content of anatase; I_A_ and I_R_ signified the integral intensities of anatase (101), and rutile (110) peaks. 

Table 2 shows the calculation results. The content of anatase TiO_2_ and rutile TiO_2_ in the pristine TiO_2_ nanofiber was 11.5 and 88.5%, respectively. With the increasing Ni doping content, their ratio was first increased, and then decreased. A maximum of 1.6247 was acquired in the sample with 3 wt.% Ni, in which the content of anatase TiO_2_ reached the highest value of about 62 wt.%. The solid solution of Ni in TiO_2_ was responsible for the increase in content of anatase TiO_2_ when the amount of Ni ranged from 0 wt.% to 3 wt.%. The Ni atoms that were included in the lattice can cause lattice distortion, thus resulting in stress field [48]. The phase transformation from anatase TiO_2_ into rutile TiO_2_ can be effectively hindered, owing to the atomic diffusion being greatly suppressed in this process. The change in NiO content in the nanofibers was the other reason causing the increase in anatase content. The transformation from anatase into rutile belonged to a process of atomic diffusion and reconstruction. The synthesized TiO particles can retard the process, which will improve the stability of anatase to a certain extent. The content of NiO presented an upward tendency in the nanofibers when the doping content of Ni was increased from 0 wt.% to 3 wt.%, resulting in the gradual increase in transformation difficulty from anatase into rutile. Correspondingly, an increasing amount of anatase was retained with the increase in Ni doping content (0 wt.%–3 wt.%). However, when the doping content of Ni exceeded 3 wt%, the content of NiO in the nanofibers presented a downward tendency with the increase in Ni doping content. At the same time, the ratio of anatase to rutile was also reduced. These may be because the formation of NiTiO_3_, which would consume NiO and TiO_2_. Thermodynamic calculations estimated the changes in standard Gibbs free energy (ΔG^θ^) of the following reactions.
(2)$${\mathrm{TiO}}_{2}\left(\mathrm{anatase}\right)\;+\;\mathrm{NiO}\;=\;\mathrm{NiTi}O_{3}$$
(3)$${\mathrm{TiO}}_{2}\left(\mathrm{rutile}\right)\;+\;\mathrm{NiO}\;=\;\mathrm{NiTi}O_{3}$$

All of the ΔG^θ^ values were negative when the temperature ranged from 298 to 1700 K, implying that the two reactions can spontaneously occur, as shown in Figure 3. However, anatase TiO_2_ will preferentially react with NiO in thermodynamics when compared to rutile TiO_2_, owing to the more negative ΔG^θ^ values in Reaction (2). These results can somehow support the experimental finding regarding anatase being consumed by Reaction (2).

As far as the four compounds were concerned, anatase TiO_2_ with a relatively high lithium insertion capacity (335 mAh·g^−1^) at room temperature and NiO with a high specific capacity (718 mAh·g^−1^) will play leading roles in improving the capacity of lithium ion batteries. Therefore, the optimal Ni doping content can be confirmed to be about 3 wt.%, in which the sample might exhibit the best electrochemical performance.

XPS analysis was performed on samples with Ni content set at 3 and 6 wt.% (Figure 4). The narrow and sharp peaks in the XPS spectra confirmed that both of the samples contained Ti, Ni, O, and C (Figsure 4a and b). The sharp peak of C1s can be abscribed to the adventitious carbon-based contaminant from the XPS instrument itself or from the sample preparation [49]. The peaks that were related to Ni were very weak in the 3 wt.% Ni-doped TiO_2_ nanofibers, owing to the comparatively low content of Ni. When the amount of Ni was further increased to 6 wt.%, the peak can be clearly identified. For the O1s spectra (Figure 4c,d), a very strong peak can be clearly observed at about 529 eV, thus confirming the existence of oxygen ions with a charge of minus two [50]. Besides that, another weak peak appeared at about 532 eV, demonstrating a significant downward trend in intensity with the increase in Ni doping content. This peak was attributed to the absorbed oxygen, which existed in the form of absorbed water or surface hydroxyl groups [49]. The contents of lattice oxygen and absorbed oxygen were calculated to be approximately 68.96 and 31.04 wt.%, respectively, in the 3 wt.% Ni-doped TiO_2_ nanofibers. However, the content of lattice oxygen was rapidly enhanced to 93.17 wt.% in the 6 wt.% Ni-doped sample, being accompanied by a decrease to 6.83 wt.% in content of absorbed oxygen. The increase in Ni doping content was responsible for this phenomenon. When Ni^2+^ was substituted for Ti^4+^ in the TiO_2_ lattice, a portion of absorbed oxygen was transformed into lattice oxygen because of the increase in number of the oxygen vacancies, resulting in a sharp decrease in the absorbed oxygen content in the sample with a higher Ni doping content. Figure 4e,f showed the Ti 2p spectra. Two peaks that are related to Ti 2p_1/2_ (about 464 eV) and Ti 2p_3/2_ (458 eV) can be well fitted, proving the existence of TiO_2_. Figure 4g showed the high-resolution scanning over the Ni 2p_3/2_ peak in the 3 wt.% Ni-doped sample. A large number of weak and cluttered peaks appeared, owing to a comparatively low Ni content. However, a comparatively strong peak that was associated with Ni 2p_3/2_ can be vaguely distinguished. When the content of Ni was increased to 6 wt.% (Figure 4h), two pairs of peaks can be clearly observed, corresponding to a Ni 2p_3/2_ peak (situated at around 855.38 eV) accompanied by a satellite peak located at around 862.88 eV, and a Ni 2p_1/2_ peak located at around 873.8 eV accompanied by a satellite peak located at around 879.1 eV. The fitted results confirmed the presence of NiO. The above results further validated that Ni was doped into TiO_2_ in the solute and compound forms.

Figure 5 shows the Raman spectra of pristine TiO_2_, as well as 3 wt.% and 6 wt.% Ni-doped TiO_2_ at room temperature. The spectra demonstrated five Raman active modes that were associated with the anatase TiO_2_ lattice [51]: E_g(1)_ (146.0 cm^−1^), E_g(2)_ (199.5 cm^−1^), B_1g_ (1) (399.8 cm^−1^), A_1g_ + B_1g_ (2) (519.2 cm^−1^), and E_g(4)_ (642.8 cm^−1^). By contrast, rutile had two Raman active modes: E_g(3)_ (447.0 cm^−1^) and A_1g(2)_ (612.0 cm^−1^). Only one peak (E_g(1)_) corresponded to anatase, and the other two peaks (E_g(3)_ and A_1g(2)_) were typical peaks of rutile in the pristine TiO_2_ nanofibers. Four characteristic peaks (E_g(2)_, B_1g_ (1), A_1g_+B_1g_ (2), and E_g(4)_) appeared in terms of the Ni-doped TiO_2_ nanofibers, which all belonged to anatase TiO_2_. Furthermore, when the Ni doping content was 3 wt.%, the peaks that were relate to anatase were the strongest, while the peaks related to rutile were the weakest, which indicated that anatase was the main phase in 3 wt.% Ni-doped TiO_2_ nanofibers. However, when the Ni doping content reached 6 wt.%, the anatase peaks became lower, while the rutile peaks appeared again. In addition, widening of Raman peaks was observed when the Ni doping content was increased. Therefore, the Raman spectra results were consistent with the XRD results. No peaks belonging to Ni were detected, since the Ni doping content was very small. However, according to the changes in other peaks, it can be proven that Ni was successfully doped into the TiO_2_ nanofibers. The conversion of anatase to rutile TiO_2_ was hindered by introducing Ni due to the disruption and the rebuilding of the original TiO_2_ structure. The TiO_2_ lattice was distorted and some oxygen vacancies were formed to maintain the electric neutrality, owing to Ti^4+^ substituted for Ni^2+^ [52], which also confirmed that Ni^2+^ was mainly doped into the TiO_2_ lattice by substitution for Ti^4+^.

Field-emission scanning electron microscope (FE-SEM) examined detailed morphologies of the samples. All of the samples presented a continuous one-dimensional nanostructure with an average diameter of 120 ± 20 nm, as shown in Figure 6 (Figure 6a,c,e). It can be confirmed that the uniform TiO_2_ nanofibers were prepared by electrospinning accompanied with annealing. The pristine TiO_2_ nanofibers comprised a large number of comparatively coarse nanograins with a diameter of around 45 nm (Figure 6b). When the Ni doping content was 3 wt.%, the nanofibers were packed with numerous finer nanograins with a diameter of about 25 nm (Figure 6d). The nanofibers’ surfaces also became rougher due to the increase in the number of grain boundaries. The nanofibers’ surfaces with 6 wt.% Ni exhibited a relatively smooth structure (Figure 6f), similar to those without Ni doping. However, the grains were obviously refined. This evolution in morphology can be attributed to the change in weight fraction of anatase TiO_2_ and rutile TiO_2_. The weight fractions were similar in the TiO_2_ nanofibers without and with 6 wt.% Ni doping (10–20% for anatase TiO_2_ and 80–90% for rutile TiO_2_), resulting from a similar surface morphology. However, the grains of the latter were finer than those of the former, because of NiO located at the grain boundaries retarding the growth of TiO_2_ grains, as well as the formation of NiTiO_3_ swallowing a portion of TiO_2_ grains. When 3 wt.% Ni was doped, the weight fraction of rutile TiO_2_ and anatase TiO_2_ was reversed (60% for anatase TiO_2_ and 40% for rutile TiO_2_), and the surface morphology of TiO_2_ nanofibers demonstrated a significant change. Fine grains were still related to the formation of NiO.

Figure 7 represented the TEM results of pristine TiO_2_, 3 wt.% and 6 wt.% Ni-doped TiO_2_ nanofibers. Nanograins can be clearly observed in all of the samples, which was consistent with the FE-SEM results. The HRTEM images revealed that all of the samples demonstrated good crystallinity, in which a large number of ordered lattice fringes with different orientations could be observed. The interplanar spaces of 0.3520 nm and 0.2970 nm can be confirmed to match those of anatase TiO_2_ (101) plane and rutile TiO_2_ (200) plane, respectively, as shown in Figure 7d. For the 3 wt.% Ni-doped TiO_2_ nanofibers (Figure 7e), in addition to the lattice fringes related to rutile TiO_2_ and anatase TiO_2_, a new interplanar space of 0.2404 nm was confirmed to agree well with that of the NiO (021) plane. From Figure 7f, besides the planes from anatase TiO_2_, rutile TiO_2_, and NiO, an extra interplanar space of 0.2528 nm can be indexed for the 6 wt.% Ni-doped TiO_2_ nanofibers, which was aligned with that of the NiTiO_3_ (111) plane. The corresponding selected-area electronic diffraction (SAED) patterns were taken from the specific zones of three samples (Figure 7g–i). A series of discontinuous concentric diffraction rings were observed. By calculation and comparison with the d values of JCPDS cards, planes that were related to rutile TiO_2_ and anatase TiO_2_ can be identified in each sample. NiO was confirmed to be synthesized in the samples with 3 wt.% and 6 wt.% Ni-doped TiO_2_; NiTiO_3_ was also formed, coupled with NiO, when the doping content of Ni was further increased to 6 wt.%. The HRTEM and SAED results agreed with those that were obtained in XRD.

The EDS mapping images of Ti and Ni in the samples with 3 wt.% and 6 wt.% Ni-doped TiO_2_ were also examined (Figure 8). The nanofibers were rich in Ti, and Ti elements were uniformly distributed throughout the whole nanofibers. Ni elements were also detected in the two nanofibers, and their distribution was relatively homogeneous. Comparatively speaking, the samples that were enriched in Ni showed more nickel (Figure 8f). The chemical compositions in the microzones were also detected by the EDS point analyses (Figure 9). The results showed that the actual contents of Ni in the samples with the theoretical doping contents of 3 wt.% and 6 wt.% were approximately 2.5 wt.% and 4.7 wt.%, respectively.

### 3.2. Electrochemical Performance of Nanofibers

To fully comprehend the influence of Ni doping content on the electrochemical performance of TiO_2_ nanofibers, the electrochemical performances of Ni-doped TiO_2_ nanofibers were investigated by assembling CR2032 coincells with lithium foils the counter electrode. Cyclic voltammetry (CV) and galvanostatic charge/discharge tests were conducted. The CV test was carried out on an electrochemical workstation with the voltage range of 0–3 V (vs. Li^+^/Li) and a sweeping rate of 0.2 mV⋅s^−1^. Figure 10a showed the cyclic voltammograms (CV) of TiO_2_ nanofibers with different Ni contents. For the pristine TiO_2_ nanofibers, a pair of characteristic redox peaks at 2.24 and 1.6 V corresponded to the extraction and insertion of Li^+^ from anatase TiO_2_ [53]. During charging, lithium ions are provided with sufficient energy by an external electric field, and are then de-intercalated from the cathode material into the electrolyte and embedded into the anode material through the separator and the electrolyte. During discharging, lithium ions are deintercalated from the anode material into the electrolyte under the action of an external electric field, and then re-embedded into the cathode material through the separator. The correspondingly reversible reactions were followed:(4)$$TiO_{2}+xLi^{+}+xe^{-}↔Li_{x}TiO_{2}$$
where x was the Li^+^ deintercalation/intercalation coefficient (in terms of anatase TiO_2_). In general, for first extraction and all other subsequent stages, the value of x did not exceed 0.5, but it might be more than 0.5 for first Li^+^ insertion process [54]. 

When the amount of Ni was 3 wt.%, a pair of additional peaks were clearly observed at 1.8 V vs Li^+^/Li and 1.3 V vs Li^+^/Li besides the above-mentioned redox peaks. While considering the change in phase constituents of TiO_2_ nanofibers with the addition content of Ni, the appearance of additional redox peaks can be ascribed to the reactions that occur between NiO and Li^+^, as follows:(5)$$NiO+2Li^{+}+2e^{-}↔Ni+Li_{2}O$$

It was worth noting that the integrated area surrounded by the CV curve recorded on the 3 wt.% Ni doped TiO_2_ nanofibers was much higher than that obtained on the pristine TiO_2_ nanofibers, which implies that the capacity of TiO_2_ nanofibers can be significantly improved by introducing NiO. The discharge capacity of NiO was more than twice than that of TiO_2_ (theoretical specific capacity: 718 mAh·g^−1^ for NiO, 335 mAh·g^−1^ for TiO_2_). Two pairs of similar redox peaks were detected when the Ni doping content was further increased to 3.5 wt.% and 4.0 wt.%; however, the integrated area surrounding the CV profiles presented a downward trend. This phenomenon might derive from the formation of NiTiO_3_, which consumed a portion of anatase TiO_2_ and NiO. The optimal Ni doping content into TiO_2_ can be confirmed as 3 wt.%, at which the excellent electrochemical performance was expected. Figure 10b compared the representative CVs of 3 wt.% Ni-doped TiO_2_ nanofibers in the first three cycles at a scanning rate of 0.2 mV·s^−1^ in the voltage range of 0-3 V (vs Li^+^/Li). The curves of the second and third cycle had a high coincidence degree, which indicated that the 3 wt.% Ni-doped TiO_2_ nanofibers had good structural stability and electrochemical reversibility. 

Figure 10c was the first cycle charge/discharge profiles obtained for the TiO_2_ nanofibers with different Ni doping contents at a current density of 100 mA·g^−1^ in a voltage window from 0 to 3.0 V. The profiles of pristine TiO_2_ nanofibers exhibited two short voltage platforms that were situated at 1.6 and 2.3 V during the Li^+^ insertion (discharge) and Li^+^ extraction (charge), which were in accordance with the redox peaks that were observed in the CV curves. The first discharge and charge capacities were 289 and 103 mAh·g^−1^, respectively, with an initial coulombic efficiency of only 35.5%. This low initial coulombic efficient was closely related to the SEI film that formed on the surface of active materials resulting from reactions occurring between active substances and electrolytes. Considerable lithium ions participated in the formation of SEI film, which was involved in the process of Li^+^ intercalation. However, the lithium ions that were confined in the SEI layer were difficult to be released in the subsequent Li^+^ deintercalation process, which causes a sharp reduction in charge capacity [55]. In addition, one more irreversible process, namely the trapping of Li ions within the lattices of anatase and rutile TiO_2_, can be considered as the other essential factor causing the decrease in coulombic efficiency. The process of Li ions embedded in the TiO_2_ lattices will inevitably cause the change in lattice. After the delithiation reaction, a portion of Li ions were not released and fixed inside TiO_2_, resulting in the low coulombic efficiency during the first charge and discharge [53]. The other researches on lithium ion batteries using TiO_2_ as an anode material had confirmed that [54,55]. When Ni was doped into TiO_2_, the second wide platform can be observed at about 1.3 V during discharge, which was regarded as the result of NiO reduction, as mentioned in the CV results. NiO demonstrated a stronger ability to intercalate Li^+^ than anatase TiO_2_ owing to the wider voltage platform caused by the former. The width of the voltage platform also strongly depended on the Ni doping content, which was first increased, and then gradually reduced. The largest width can be acquired at the Ni doping content of 3 wt.% because of the highest content of NiO being synthesized. No extra platforms appeared during the charge, which may result from approximately equivalent Li^+^ deintercalation potentials for anatase TiO_2_ and NiO. The discharge and charge specific capacity was also increased from 479/193 to 576/264 mAh·g^−1^ as the Ni doping content was increased from 1 wt.% to 3 wt.%. However, as the Ni doping content continued to be increased to 3.5 wt.% and 4 wt.%, the discharge/charge specific capacity dropped to 436/195 and 378/120 mAh·g^−1^. Similarly, the coulombic efficiency was also first increased and then decreased (35.5, 40.3, 45, 44.6, and 31.7% for 0, 1, 3.5, and 4 wt.% Ni-doped TiO_2_). As far as the coulombic efficiency was concerned, some investigations confirmed that the coulombic efficiency of the NiO electrode material (~80%) [56] was generally better than that of the pristine TiO_2_ electrode material (~50%) [43]. Therefore, a similar change in coulomb efficiency was also observed in the TiO_2_ nanofibers with different doping contents of Ni. Figure 10d showed the charge-discharge specific capacity of the 3 wt.% Ni-doped TiO_2_ nanofibers at different cycles. The discharge/charge capacity demonstrated a declining trend, along with an increase in cycles. However, the charge-discharge curves gradually tended to coincide with each other after the 25th cycle.

The evolution in coulombic efficiency and discharge capacity of the 3 wt.% Ni-doped TiO_2_ nanofibers with respect to the cycling number was calculated and compared with those of the other Ni-doped nanofibers (Figure 10e). The pristine TiO_2_ nanofibers presented a relatively low specific discharge capacity of 45 mAh·g^−1^ after 100 cycles (about 36% initial discharge capacity was retained). After 100 cycles, the value was improved to 84.3 and 123.5 mAh·g^−1^ when the Ni doping content was increased to 1 wt.% and 3 wt.%, respectively. Their capacity retention rates were correspondingly evaluated as 46% and 48%. However, the discharge capacity began to fall back with further increasing the Ni doping content (79.8 and 48.2 mAh·g^−1^ for the 3.5 and 4 wt.% Ni-doped TiO_2_ nanofibers), while the capacity retention rates were also reduced to 41% and 38%, respectively. Based on the above analyses, it can be summarized, as follows: suitable Ni doping into TiO_2_ nanofibers (3 wt.%) can greatly improve the cycling stability of TiO_2_ nanofibers, namely the discharge capacity was nearly doubled.

Rate capability was evaluated as current density was increased from to 40, 100, 200, 400, and 1000 mA·g^−1^ in turn, and finally recovered to 40 mA·g^−1^ to acquire the kinetics of pristine TiO_2_ and four kinds of Ni-doped TiO_2_ anodes. The value was measured six times at every given current density. The average specific discharge capacity of the 3 wt.% Ni-doped TiO_2_ nanofibers was generally higher than those of the other nanofibers at each current density (219, 131, 97, 74, 48, and 144 mAh·g^−1^ at 40, 100, 200, 400, 1000, and 40 mA·g^−1^), as shown in Figure 10f. Accompanied by the current density increased from 40 to 1000 mA·g^−1^, the average specific discharge capacity of this material was decreased from 219 to 48 mAh·g^−1^, with a higher retention rate (22%) than those of the other materials (13.5, 17.9, 21, and 10.3% for 0, 1, 3.5, and 4 wt.% Ni-doped TiO_2_). When the current density returned to 40 mA·g^−1^, ~65.8% of the initial value was retained (144 mAh·g^−1^), which was slightly higher than those that were retained in the other materials (62.5, 62.7, 65.2, and 65.0% for 0, 1, 3.5, and 4 wt.% Ni-doped TiO_2_). This confirmed that the TiO_2_ nanofibers with 3 wt.% Ni possessed the optimal rate capability. In addition, a similar phenomenon was observed, namely that the rate capability was first increased, and then decreased with increasing the Ni doping content.

The above results revealed that the TiO_2_ nanofibers can be endowed with excellent electrochemical performance with respect to specific discharge capacity, cycling stability, and rate capability by introducing 3 wt.% Ni into TiO_2_ nanofibers. This was due to the two significant changes that were caused by doping 3 wt.% Ni into TiO_2_, namely 1) the solution content of Ni embedded in the TiO_2_ lattice reached the maximum value; and, 2) the content of NiO that was involved in TiO_2_ reached the highest value. The first change was associated with the largest lattice distortion of TiO_2_, resulting in the inhibition of transformation from anatase into rutile. The second one was related to the large number of uniformly distributed NiO particles retarding the atomic diffusion and recombination [48], which in turn reduced the nucleation and growth rates of anatase, and intensely suppressed its transformation into rutile. Therefore, the highest content of fine anatase was involved in the TiO_2_ nanofibers with 3 wt.% Ni doping. The above change in structure of TiO_2_ will greatly improve the electrochemical performance of the material. Anatase TiO_2_ belongs to a tetragonal system with the unit cell parameters of a = 37.9 nm, b = 95.1 nm, which is formed by sharing four sides and apexes that are based on the octahedron of TiO_6_. There are bidirectional pore channels along the a-axis and b-axis, which allow for a larger number of lithium ions to transport and insert in the lattice of anatase TiO_2_ at room temperature [57]. As a result, a high insertion capacity can be observed. Rutile TiO_2_ belongs to a tetragonal system with unit cell parameters of a = 45.9 nm, c = 29.6 nm. Although the TiO_6_ coordination octahedron is joined by the co-edges to form chains that are parallel to the c-axis, the pore channel along the c-axis is narrow, and the radius of the octahedral vacancy is 4 nm less than that of the lithium ion (6 nm). This crystal structure makes transporting and embedding of lithium ions into rutile TiO_2_ very difficult at room temperature. Therefore, anatase TiO_2_ is superior to rutile TiO_2_ in terms of electrochemical performance because of its higher conductivity for lithium ions and electron transportation, and more vacancies for lithium ion insertion. Some studies have verified this. Jiang el al. [58] investigated the electrochemical performance of anatase TiO_2_ nanoparticles (6 nm) in 1 mol/L LiCLO_4_/EC+DMC (1:1, v/v) electrolyte; at 0.1 A·g^−1^, the specific discharge capacity was 234 mAh·g^−1^. Baudrin el al. [59] prepared 50 nm of rod-shaped rutile TiO_2_ particles with a comparatively low initial discharge specific capacity of only 77 mAh·g^−1^ at a current density of 50 mA⋅g^−1^. Obviously, the highest content of synthesized anatase in the nanofibers with 3 wt.% Ni doping made an important contribution to the improvement in the electrochemical performance of TiO_2_ nanofibers. Moreover, the grain refinement in anatase can be regarded as the second important factor that is responsible for the improvement in the electrochemical performance of TiO_2_ nanofibers due to the inerparticle contact in TiO_2_ strengthened and the diffusion paths of Li^+^ shortened [37,38]. In addition, the presence of the highest content of NiO was another essential factor for the obtained excellent electrochemical performance in the nanofibers with 3 wt.% Ni doping, owing to its higher theoretical specific capacity (718 mAh⋅g^−1^) by participating in reversible reactions and charge storage ($NiO+2Li^{+}+2e^{-}↔Ni+Li_{2}O$) [56]. Vanchiappan Aravindan et al. [60] synthesized one-dimensional NiO nanofibers by electrospinning, followed by annealing at 800 °C, which demonstrated a maximum reversible capacity of 784 mAh⋅g^−1^ at a current density of 80 mA⋅g^−1^. Sun et al. [61] used the thermal oxidation process to produce three-dimensionally “curved” NiO nanomembranes, which showed a high capacity of 721 mAh⋅g^−1^ at 1.5 C (1C = 718 mAh⋅g^−1^). When the Ni doping content exceeded 3 wt.%, the contents of anatase TiO_2_ and NiO were gradually decreased because the formation of NiTiO_3_ consumes NiO and anatase TiO_2_, which caused the corresponding reduction in electrochemical performance. The optimal electrochemical performance can be obtained in the TiO_2_ nanofibers with 3 wt.% Ni. 

The electrochemical performance of the 3 wt.% Ni-doped TiO_2_ nanofibers electrode was compared with that of other reported nanostructured TiO_2_ composites (as shown in Table 3). The first discharge/charge capacity and cycle stability values in our study were generally higher than those that were reported in the other references.

Figure 11 shows the EIS and fitting equivalent circuit of the pristine TiO_2_ and different Ni-doped TiO_2_ nanofibers. All the nanofibers’ EIS curves contained a semicircle in the high frequency region and a straight line (Warburg tail) in the low frequency region. The intercept of the semicircle with the real axis can be used to characterize the transfer resistance of the electrolyte to the electrode surface (R_s_). The charge-transfer resistance (R_ct_) can be characterized by the radius of the high frequency semicircle, and the impedance lithium ions diffusion in the electrode material (Warburg impedance) depended on the slope of the straight line [64,65]. It can be clearly illustrated from the partial enlargement that the R_s_ values of all the samples were comparatively low (less than 5.0 Ω). The R_ct_ value of the pristine TiO_2_ nanofibers was 158.7 Ω, which was gradually decreased to 154.8 and 90.6 Ω with the increase to 1 and 3 wt.% in Ni doping content. However, the value was increased from 153.3 to 182.7 Ω as the Ni doping content was further increased to 3.5 and 4 wt.%. The conductivity of all samples can be calculated based on the EIS data [37], which demonstrated a trend of first rising and then falling (when the Ni doping content is 0, 1, 3, 3.5, and 4 wt.%, the corresponding conductivity was 2.82 × 10^−5^, 2.89 × 10^−5^, 4.92 × 10^−5^, 2.93 × 10^−5^, and 2.46 × 10^−5^, respectively). The Warburg impedance also presented a similar change with respect to the Ni doping content. It can be clearly seen that the lithium ions can diffuse and participate in the redox reactions at a faster speed in the 3 wt.% Ni-doped TiO_2_ nanofibers when compared to the other nanofibers. This should be attributed to the highest content of anatase TiO_2_, which was synthesized in the 3 wt.% Ni-doped TiO_2_ nanofibers. The highest content of anatase was synthesized in the 3 wt.% Ni-doped TiO_2_ nanofibers, accompanied by which the lowest content of rutile was obtained, as mentioned above. Anatase TiO_2_ is formed by sharing four sides and apexes that are based on the octahedron of TiO_6_, and these bidirectional channels along the a-axes and b-axes provide two channels for lithium ion migration. When compared with anatase, the rutile TiO_6_ octahedron only forms a chain parallel to the c axis by co-edge connection, which only provides a channel for lithium ion migration. Moreover, the pore channel with a radius of only 4 nm is narrower, which only allows for a small amount of lithium ions to be inserted [57]. Therefore, the 3 wt.% Ni-doped TiO_2_ nanofibers with a highest content of anatase can provide more channels with a larger radius for lithium ion transportation and the insertion/extraction of lithium ions. 

## 4. Conclusions

We successfully synthesized uniform and stable Ni-doped TiO_2_ nanofibers by the electrospinning method followed by annealing. The applied synthesized method was simple and easy to operate, and the doped NiO particles were uniformly distributed throughout the whole TiO_2_ nanofibers. This mixed structure was different from those that were prepared in past investigations, which can retard the transformation from anatase into rutile and fully exploit the synergistic effect between TiO_2_ and NiO. The advantages were also supported by the following results:

The XRD experiments showed that the doping of Ni played a decisive role in the phase transition of TiO_2_. When the doping content of Ni was increased from 0 to 3 wt.%, the ratio of anatase to rutile was increased, a maximum of ratio of 1.6247 was acquired in the 3 wt.% Ni-doped TiO_2_ nanofibers, at this time, the content of NiO also reached a maximum. When the doping content of Ni exceeds 3 wt.%, due to the formation of NiTiO_3_, NiO and anatase were consumed, which caused the decrease in content of NiO and anatase. At the same time, the content of rutile was correspondingly increased.

The electrochemical performance test results showed that 3 wt% Ni-doped TiO_2_ nanofibers had the best electrochemical performance. When the Ni doping content was 3 wt.%, it had a significantly higher initial discharge capacity than pristine TiO_2_ (576.8 vs 289 mAh·g^−1^). The excellent cycle stability was also obtained after 100 cycles at 100 mA·g^−1^, 48% of the initial discharge capacity was retained^,^, which was higher than those of the other samples. When the current density was increased from 40 to 1000 mA·g^−1^, 3 wt.% Ni-doped TiO_2_ nanofibers retained 22% of the initial discharge capacity, when the current density was returned to 40 mA·g^−1^, 65.8% of the initial value was retained. 

Based on the experimental results, the optimal doping amount of Ni was determined as 3 wt.%, in which the Ni-doped TiO_2_ nanofibers demonstrated the better comprehensive electrochemical performance than those that were synthesized by the other methods. The 3 wt.% Ni-doped TiO_2_ nanofibers can be regarded as an extremely promising anode material for lithium ion batteries. The subsequent investigation should focus on further improvement in electrochemical performance by more uniformly dispersing NiO particles within TiO_2_ nanofibers by means of adjusting and optimizing the operating parameters.

## Figures and Tables

**Figure 1 materials-13-01302-f001:**
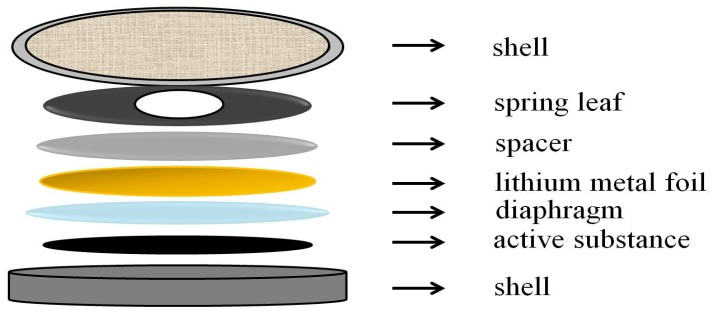
The diagram of the cell configuration.

**Figure 2 materials-13-01302-f002:**
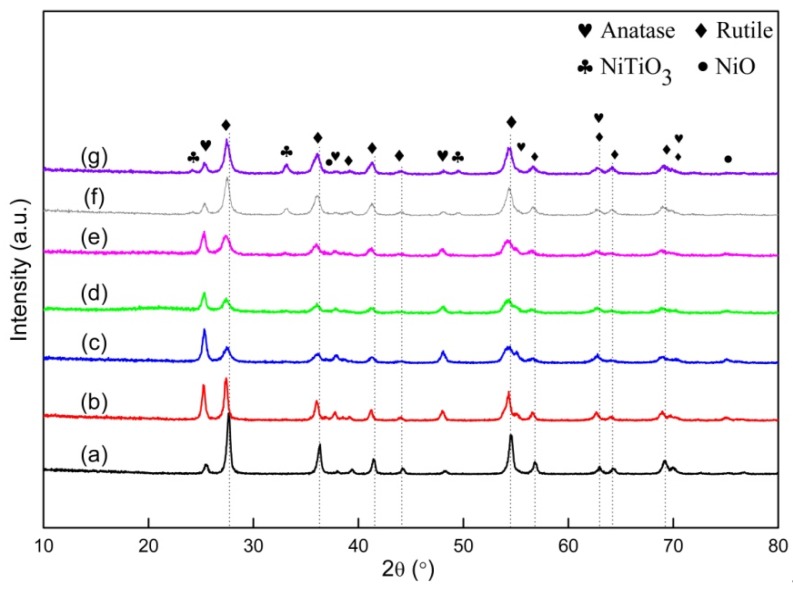
XRD patterns of (**a**) pristine TiO_2_, (**b**) 1 wt.% Ni-doped TiO_2_, (**c**) 3 wt.% Ni-doped TiO_2_, (**d**) 3.5 wt.% Ni-doped TiO_2_, (**e**) 4 wt.% Ni-doped TiO_2_, (**f**) 6 wt.% Ni-doped TiO_2_, and (**g**) 10 wt.% Ni-doped TiO_2_.

**Figure 3 materials-13-01302-f003:**
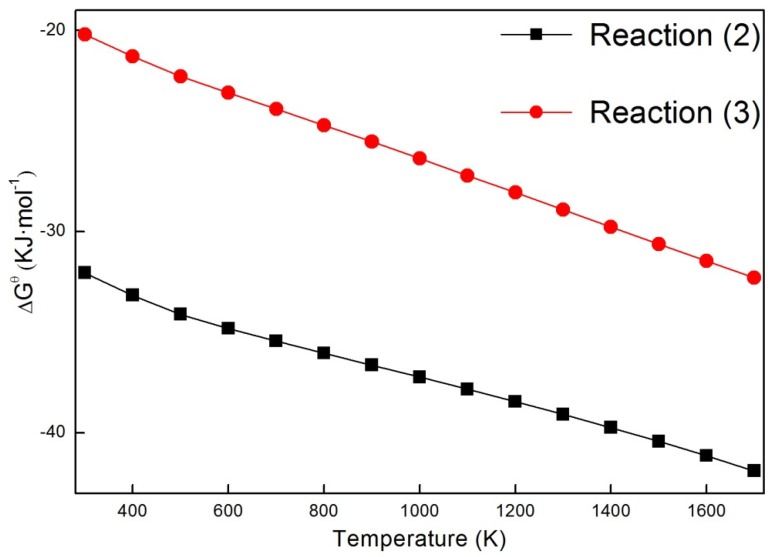
Gibbs free energy change (△G^θ^) for Reactions (2) and (3).

**Figure 4 materials-13-01302-f004:**
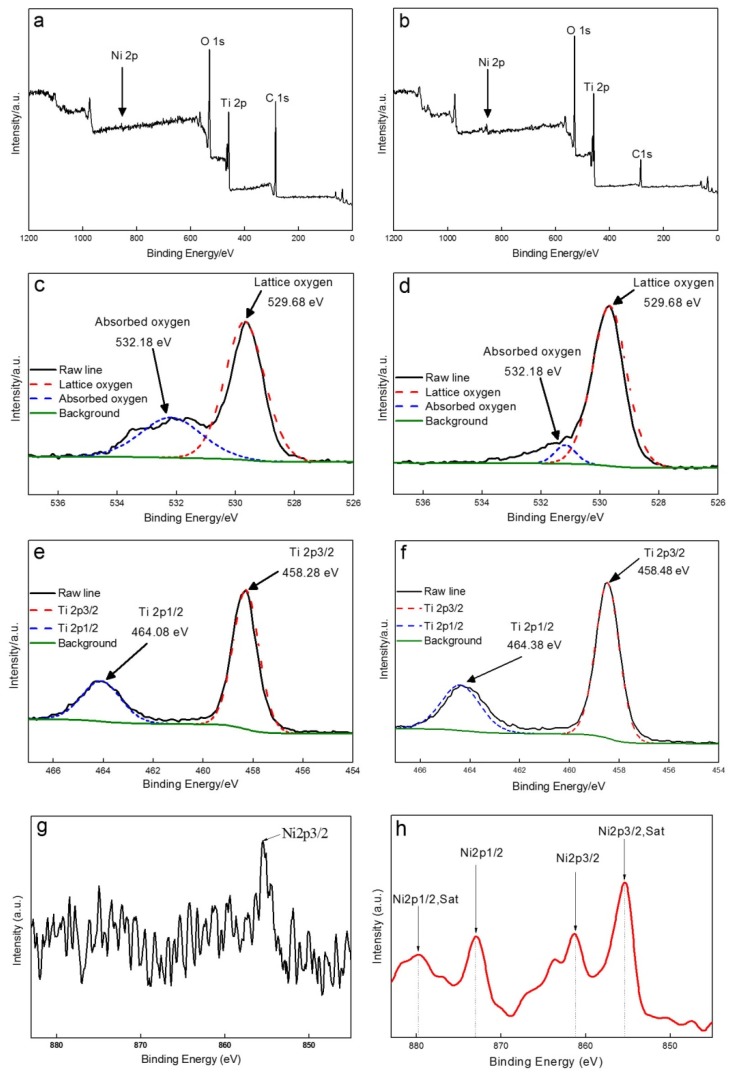
X-ray photoelectron spectroscope (XPS) survey spectra (**a**,**b**) and high-resolution XPS spectra of O 1s (**c**,**d**), Ti 2p (**e**,**f**), and Ni 2p (**g**,**h**) of the 3 wt.% Ni-doped TiO_2_ nanofibers and 6 wt.% Ni-doped TiO_2_ nanofibers.

**Figure 5 materials-13-01302-f005:**
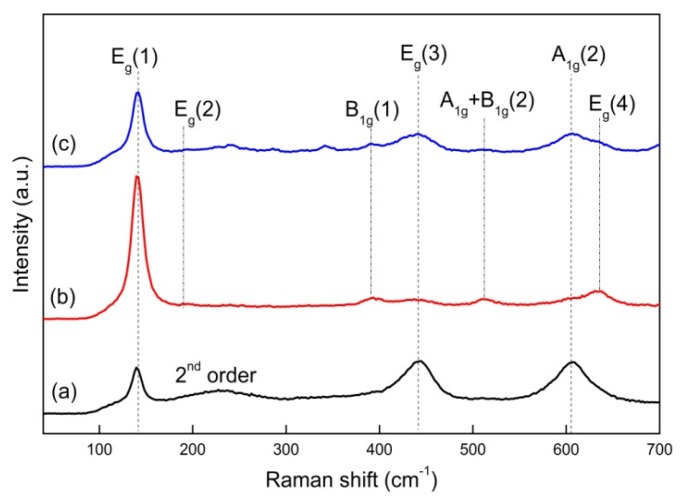
Raman spectra of (**a**) pristine TiO_2_, (**b**) 3 wt.% Ni-doped TiO_2_, and (**c**) 6 wt.% Ni-doped TiO_2_.

**Figure 6 materials-13-01302-f006:**
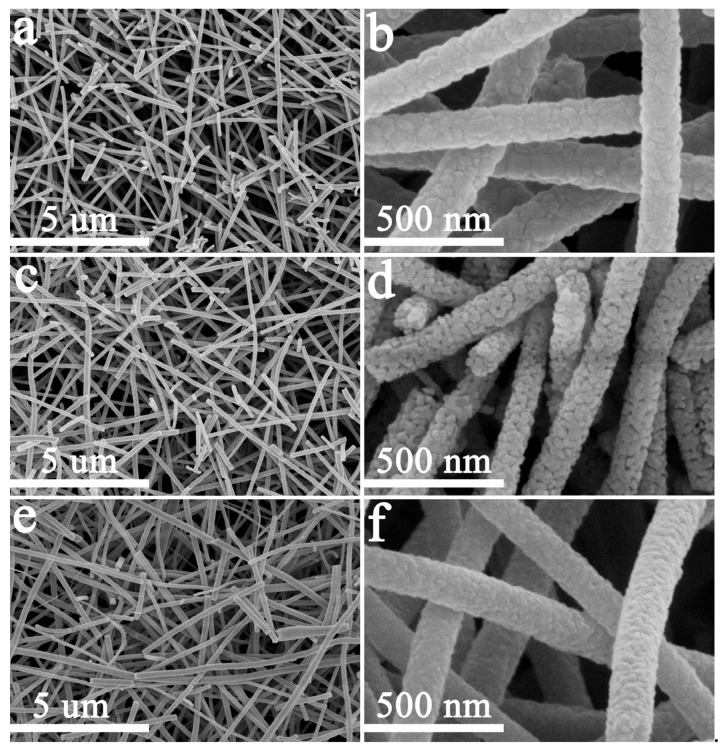
Field-emission scanning electron microscope (FE-SEM) images of (**a**,**b**) pristine TiO_2_ nanofibers; (**c**,**d**) 3 wt.% Ni-doped TiO_2_ nanofibers; and, (**e**,**f**) 6 wt.% Ni doped TiO_2_ nanofibers.

**Figure 7 materials-13-01302-f007:**
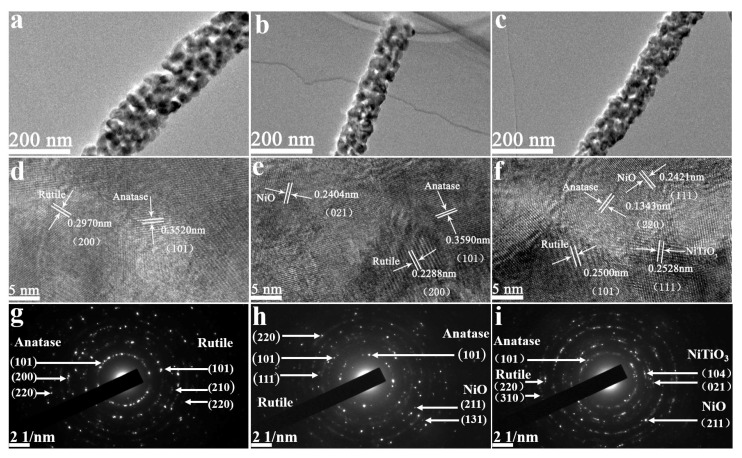
Transmission electron microscope (TEM) images of (**a**) pristine TiO_2_ nanofibers, (**b**) 3 wt.% Ni-doped TiO_2_ nanofibers, and (**c**) 6 wt.% Ni-doped TiO_2_ nanofibers. HRTEM images of (**d**) pristine TiO_2_ nanofibers, (**e**) 3 wt.% Ni-doped TiO_2_ nanofibers, (**f**) 6 wt.%. Ni-doped TiO_2_ nanofibers. SAED patterns of (**g**) pristine TiO_2_ nanofibers, (**h**) 3 wt.% Ni-doped TiO_2_ nanofibers, and (**i**) 6 wt.% Ni-doped TiO_2_ nanofibers.

**Figure 8 materials-13-01302-f008:**
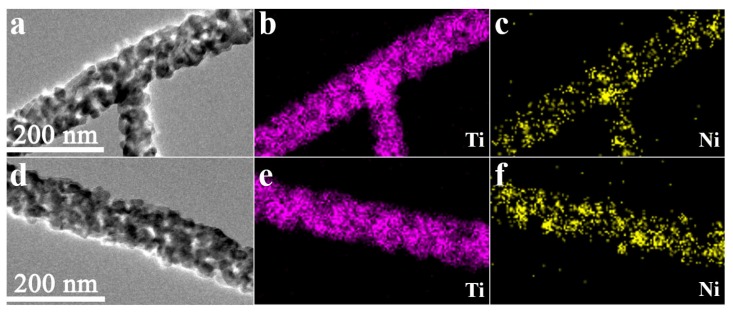
Elemental mapping images of (**a**–**c**) 3 wt.% Ni-doped TiO_2_ nanofibers and (**d**–**f**) 6 wt.% Ni-doped TiO_2_ nanofibers.

**Figure 9 materials-13-01302-f009:**
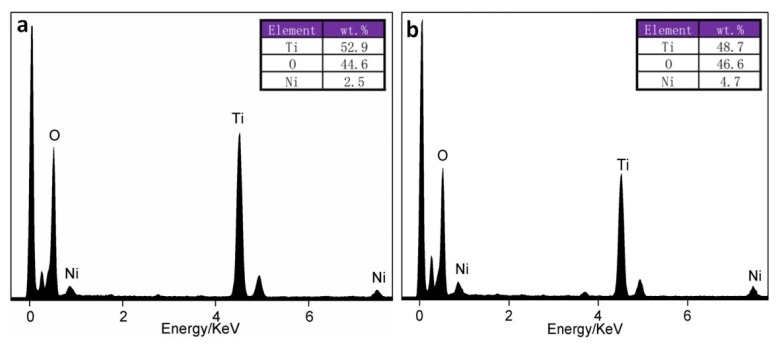
Energy-dispersive X-ray detector (EDS) point analyses of (**a**) 3 wt.% Ni-doped TiO_2_ nanofibers and (**b**) 6 wt.% Ni-doped TiO_2_ nanofibers.

**Figure 10 materials-13-01302-f010:**
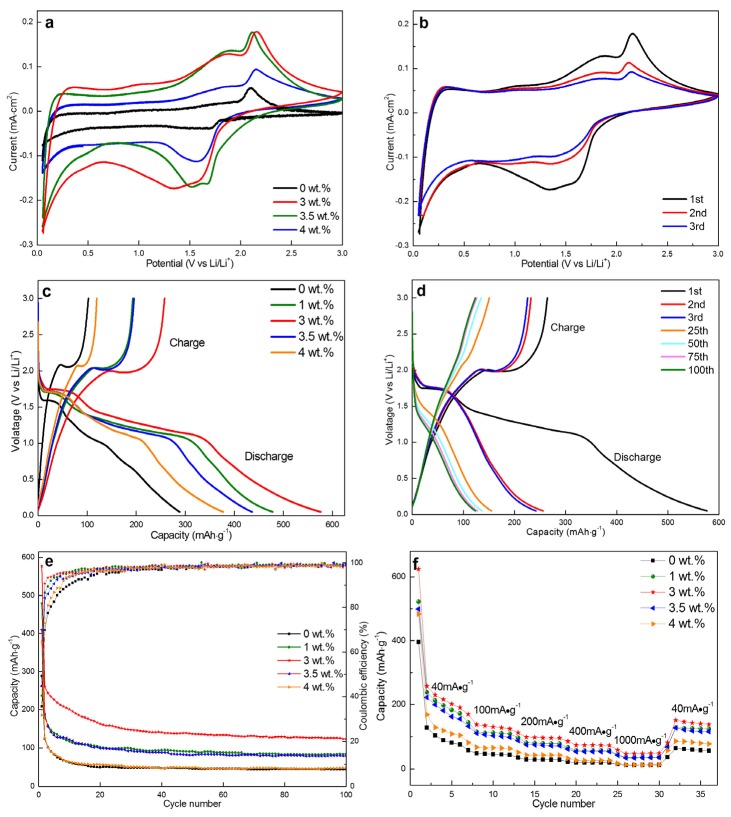
Cyclic voltammograms of (**a**) Ni-doped TiO_2_ nanofibers and (**b**) 3 wt.% Ni-doped TiO_2_ nanofibers from the first three cycles at a scan rate of 0.2 mV/s between 0-3 V. First charge-discharge curves of (**c**) Ni-doped TiO_2_ nanofibers and the first three charge-discharge curves of (**d**) 3 wt.% Ni-doped TiO_2_ nanofibers at 100mA·g^−1^. (**e**) Cycle performance and coulombic efficiency of Ni-doped TiO_2_ nanofibers at 100 mA·g^−1^. (**f**) Rate capability of Ni-doped TiO_2_ nanofibers at different current densities.

**Figure 11 materials-13-01302-f011:**
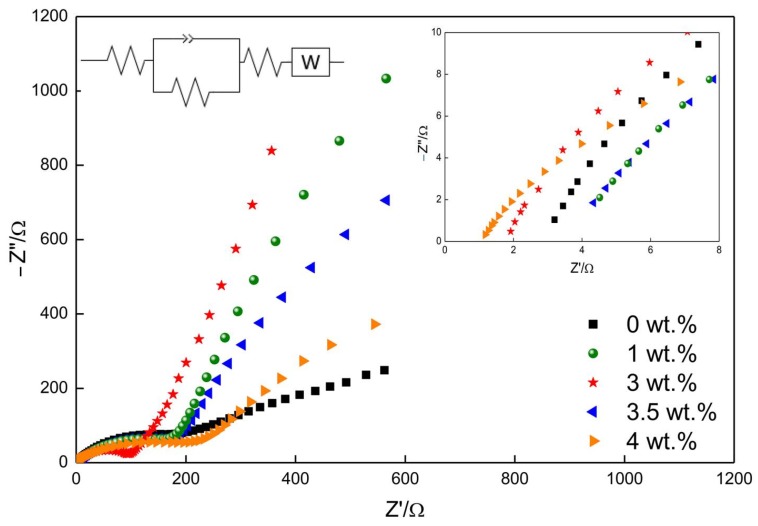
Impedance spectra of Ni-doped TiO_2_ nanofibers.

**Table 1 materials-13-01302-t001:** Refined unit-cell parameters for Ni-doped TiO_2_ nanofibers.

	Rutile				Anatase			
Sample	a/Å	b/Å	c/Å	v/Å3	a/Å	b/Å	c/Å	v/Å3
0 wt.% Ni-doped TiO_2_	4.5839	4.5839	2.9492	61.97	3.7950	3.7950	9.4568	136.20
1 wt.% Ni-doped TiO_2_	4.5885	4.5885	2.9585	62.29	3.7958	3.7958	9.4576	136.27
3 wt.% Ni-doped TiO_2_	4.5953	4.5953	2.9608	62.52	3.7910	3.7910	9.4920	136.42
3.5 wt.% Ni-doped TiO_2_	4.5953	4.5953	2.9607	62.52	3.7776	3.7776	9.5371	136.10
4 wt.% Ni-doped TiO_2_	4.5908	4.5908	2.9595	62.37	3.7845	3.7845	9.5015	136.08
6 wt.% Ni-doped TiO_2_	4.5896	4.5896	2.9576	62.30	3.7887	3.7887	9.4668	135.89
10 wt.% Ni-doped TiO_2_	4.5730	4.5730	2.9666	62.04	3.7810	3.7810	9.4681	135.36

**Table 2 materials-13-01302-t002:** Phase composition of different Ni doping content of TiO_2_ nanofibers.

Doping	Phase Composition/wt.%		Ratio of
Concentration/wt.%	Anatase	Rutile	A to R
0	11.5	88.5	0.1299
1	40.0	60.0	0.6667
3	61.9	38.1	1.6247
3.5	51.5	48.5	1.0619
4	48.5	51.5	0.9398
6	21.8	78.2	0.2788
10	21.7	78.3	0.2771

**Table 3 materials-13-01302-t003:** Electrochemical performance of nanostructured TiO_2_ composites reported in some related references.

Ref.	Materials	Discharge/Charge Capacity	Cycle Performance
Our work	3 wt.% Ni-doped TiO_2_ nanofibers	576/264 mAh·g^−1^ at 100 mA·g^−1^	124 mAh·g^−1^/100 cycles at 100 mA·g^−1^
[27]	Three dimensional TiO_2_ nanotube arrays	118.1/96 mAh·g^−1^ at 70 uA·cm^−2^	63 mAh·g^−1^/50 cycles at 100 mA·g^−1^
[37]	Nb@TiO_2_ nanofibers	252/115 mAh·g^−1^ at 16.8 mA·g^−1^	-
[38]	Zr-doped TiO_2_ nanotubes	255/150 mAh·g^−1^ at 0.1C	136 mAh·g^−1^/ 35 cycles at 0.1C
[40]	Hf-doped TiO_2_ nanofibers	321/162 mAh·g^−1^ at 33.5 mA·g^−1^	170 mAh·g^−1^/35 cycles at 33.5 mA·g^−1^
[43]	NiO doped onto TiO_2_ nanotubes	152/94 uA·cm^−2^ at 70 uA·cm^−2^	85 uA·cm^−2^/25 cycles at 70 uA·cm^−2^
[43]	Co_3_O_4_ doped onto TiO_2_ nanotubes	200.1/103 uA·cm^−2^ at 70 uA·cm^−2^	100 uA·cm^−2^/25 cycles at 70 uA·cm^−2^
[62]	Au@TiO_2_ nanofibers	180/120 mAh·g^−1^ at 66 mA·g^−1^	150 mAh·g^−1^/50 cycles at 66 mA·g^−1^
[63]	Zr^4+^/F^-^ co-doped TiO_2_ nanotubes	250/175 mAh·g^−1^ at 335 mA·g^−1^	175 mAh·g^−1^/35 cycles at 1C
[50]	10 mol% Al doped-TiO_2_ nanofibers	200/152 mAh·g^−1^ at 40 mA·g^−1^	148 mAh·g^−1^/100 cycles at 40 mA·g^−1^

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
