# Peer review of "Effect of Ni Doping Content on Phase Transition and Electrochemical Performance of TiO2 Nanofibers Prepared by Electrospinning Applied for Lithium-Ion Battery Anodes"

_materials, 2020, doi:10.3390/ma13061302_

Round 1

Reviewer 1 Report

This paper expounds the investigation of TiO2 modifying with NiO dopant through electrospinning route in order to obtain the high-performance anode for advanced Li-ion batteries. This manuscript is well organized and results are also very clear. This manuscript is recommended for publication after some revisions.

1) XRD: because of the XRD peaks are shifted the changes in lattice parameters of TiO2 may be occurred. Hence, the Rietveld refinements may be useful to study this moment.

2) What is actual concentration of Ni in samples: both EDX and/or XPS provides this information.

3) There are incorrect sentence in Lines 374-375. The peaks at 2.24 and 1.6 V correspond to extraction and insertion of Li+ ions, and not vice versa.

4) Also there are some incorrectness in Lines 377-378 for x coefficient. It may be more than 0.5 for first Li+ insertion process. Only for first extraction and all other subsequent stages it usually does not exceed 0.5.

5) When discuss about low coulombic efficiency due to SEI film (Lines 404-405), one more irreversible process, namely trapping of Li+ ions within lattice should be mentioned. Based on literature it is a typical shortcoming for both anatase and rutile.

6) The sentence «The discharge/charge specific capacity of pristine TiO2 nanofibers is 289/103 mAh g-1» (Line 416) should be deleted because of it is a repetition of the sentence in Lines 402-403.

Reviewer 2 Report

The manuscript entitled "Effect of Ni Doping Content on Phase Transition and Electrochemical Performance of TiO2 Nanofibers Prepared by Electrospinning applied for Lithium-Ion Battery Anodes" reports a very interesting and innovative work of high impact in the world of anode materials for lithium-ion batteries.
Before to accept the present manuscript the authors should revise the English language and focus the introduction section on the recent investigations on anode materials based on TiO2 for this kind of applications.

Reviewer 3 Report

My comments are attached.

Round 2

Reviewer 2 Report

All comments have been introduced in the manuscript

Author Response

Dear Reviewer,

Firstly, thanks for your careful review for my paper. Based on your suggestion, we had revised my paper of English language in detail, and the and revised parts had been marked in red, purple and blue in revised manuscript.